# Nitrous Oxide Emission from Full-Scale Anammox-Driven Wastewater Treatment Systems

**DOI:** 10.3390/life12070971

**Published:** 2022-06-28

**Authors:** Zhiman Lin, Kayan Ma, Yuchun Yang

**Affiliations:** State Key Laboratory of Biocontrol, School of Ecology, Sun Yat-sen University, Guangzhou 510275, China; linzhm25@mail2.sysu.edu.cn (Z.L.); majx26@mail.sysu.edu.cn (K.M.)

**Keywords:** nitrous oxide, anammox, wastewater treatment, mitigation

## Abstract

Wastewater treatment plants (WWTPs) are important contributors to global greenhouse gas (GHG) emissions, partly due to their huge emission of nitrous oxide (N_2_O), which has a global warming potential of 298 CO_2_ equivalents. Anaerobic ammonium-oxidizing (anammox) bacteria provide a shortcut in the nitrogen removal pathway by directly transforming ammonium and nitrite to nitrogen gas (N_2_). Due to its energy efficiency, the anammox-driven treatment has been applied worldwide for the removal of inorganic nitrogen from ammonium-rich wastewater. Although direct evidence of the metabolic production of N_2_O by anammox bacteria is lacking, the microorganisms coexisting in anammox-driven WWTPs could produce a considerable amount of N_2_O and hence affect the sustainability of wastewater treatment. Thus, N_2_O emission is still one of the downsides of anammox-driven wastewater treatment, and efforts are required to understand the mechanisms of N_2_O emission from anammox-driven WWTPs using different nitrogen removal strategies and develop effective mitigation strategies. Here, three main N_2_O production processes, namely, hydroxylamine oxidation, nitrifier denitrification, and heterotrophic denitrification, and the unique N_2_O consumption process termed *nosZ*-dominated N_2_O degradation, occurring in anammox-driven wastewater treatment systems, are summarized and discussed. The key factors influencing N_2_O emission and mitigation strategies are discussed in detail, and areas in which further research is urgently required are identified.

## 1. Introduction

Nitrous oxide (N_2_O), as a potent greenhouse gas (GHG), has a global warming potential of 298 CO_2_ equivalents [1] that contribute to the depletion of the ozone layer in the biosphere [2] and is considered the third most emitted GHG involved in global warming after carbon dioxide (CO_2_) and methane (CH_4_). Over the past decade, the atmospheric N_2_O concentration has been increasing at an average rate of ~0.31% per year [3]. A considerable proportion of N_2_O emission has occurred in domestic wastewater treatment systems, which contributed 1.6 Tg CO_2_ equivalents over the past two decades, equivalent to 1.6% of the global N_2_O emissions in 2010 [4]. It is therefore important to understand N_2_O emission mechanisms in wastewater treatment plants (WWTPs).

Anaerobic ammonia oxidation (anammox) has recently been developed as an energy-efficient way in wastewater treatment (70–90% of total nitrogen removal) [5], and over 100 anammox-processing full-scale WWTPs were implemented worldwide by 2014 [6]. Anammox bacteria provide a shortcut in the nitrogen cycle by direct transforming ammonium (NH_4_^+^) and nitrite (NO_2_^−^) to nitrogen gas (N_2_) [7], rendering this method more efficient and cost-effective than the conventional nitrification/denitrification process. Since the discovery of anammox by Mulder [8] in 1995, extensive research has been carried out to develop anammox-driven nitrogen removal technologies. Considering the limitations of the conventional wastewater treatment systems, the combination of biological processes such as in the sequencing batch reactor (SBR) stands as a promising and viable option for sewage treatment, with low cost, high efficiency, and high stability [9,10,11]. Additionally, the partial nitrification/anammox process (PNA) provides an effective new option for the treatment of high-strength NH_4_^+^ wastewater with a low C/N ratio and elevated temperature. It involves the partial oxidation of NH_4_^+^ to NO_2_^−^ and the anaerobic oxidation of the remaining NH_4_^+^ and NO_2_^−^ to N_2_. The integrated PNA process can be conducted either in independent dedicated two-stage PNA reactors separating partial nitrification from anammox or simultaneously in the same reactor where both partial nitrification and anammox occur under low dissolved oxygen (DO) conditions [12]. Early implementations of PNA systems employed a two-stage configuration for the efficient control of partial nitrification, whereas recently, the focus has turned mainly to a one-stage PNA system due to its low N_2_O emission [13,14] and operating costs [15].

Nevertheless, N_2_O emission is still one of the downsides of anammox-driven wastewater treatment. Although direct evidence of the metabolic production of N_2_O by anammox bacteria is lacking, the microorganisms coexisting in anammox-driven WWTPs, such as nitrifiers and denitrifiers, could produce a considerable amount of N_2_O and affect the sustainability of the wastewater treatment [16,17]. This work intends to offer an overview of the processes taking place during the biological production and consumption of N_2_O in anammox-driven WWTPs and to discuss the key factors influencing N_2_O emission and mitigation strategies. Potential strategies focusing on the microbial community structure in anammox-driven WWTPs deserve further investigations.

## 2. N_2_O Emission

In anammox-driven wastewater treatment systems, the net N_2_O emission is driven by four key reactions: hydroxylamine oxidation (NH_4_^+^ → NH_2_OH → N_2_O) and nitrifier denitrification (NO_2_^−^ → NO → N_2_O or NH_2_OH → N_2_O or NH_2_OH + NO → N_2_O) catalyzed by nitrifiers as well as heterotrophic denitrification (NO_3_^−^ → NO_2_^−^ → NO → N_2_O) catalyzed by diverse denitrifiers are the three known N_2_O-forming biological processes, while *n**osZ*-dominated N_2_O consumption (N_2_O → N_2_) is the unique N_2_O degradation biological process driven by denitrifiers (Figure 1).

### 2.1. Hydroxylamine Oxidation

Hydroxylamine (NH_2_OH), an inorganic and highly reactive chemical, is one of the main precursors of N_2_O production via nitrification under aerobic conditions [18]. It is produced as one of the intermediate products of the nitrification process, which begins by oxidizing ammonia (NH_3_) with ammonia monooxygenase (AMO) and particulate methane monooxygenase (pMMO) to yield NH_2_OH. Normally, NH_2_OH is then further oxidated to nitric oxide (NO) by either hydroxylamine dehydrogenase (HAO) or hydroxylamine oxidase (HOX) produced by ammonia-oxidizing archaea (AOA) and ammonia-oxidizing bacteria (AOB). The produced NO_2_^−^ from NO oxidation is then oxidated to nitrate (NO_3_^−^) by nitrite-oxidizing bacteria (NOB) using a nitrite oxidoreductase (NXR). The process can also be achieved through complete ammonia oxidation (comammox) by comammox bacteria, which encode all enzymes for complete nitrification (NH_4_^+^ → NO_2_^−^ → NO_3_^−^) [19,20].

If NH_2_OH production catalyzed by AMO and pMMO is faster than the conversion of HAO and HOX under aerobic conditions, the accumulated NH_2_OH can stimulate hydroxylamine oxidation to consume N_2_O, such that a metabolic imbalance is established [21]. The accumulated free NH_2_OH could be emitted from the cells and produce N_2_O through an abiotic chemical hybrid reaction with oxidants or extracellular NO_2_^−^, i.e., the hydroxylamine oxidation reaction [22,23], while the oxidized NO_2_^−^ can be reduced to NH_2_OH to slow down the abiotic decay of NH_2_OH [23]. Based on NH_2_OH abiotic conversion rates, the maximum proportions of NH_4_^+^ converted to N_2_O via extracellular NH_2_OH during the incubation of AOB, AOA, and comammox (*Nitrospira inopinata*) have been estimated to be 0.12%, 0.08%, and 0.14%, respectively [24]. This result is consistent with a prior study on the NH_4_^+^:N_2_O conversion ratio by AOB and AOA, which demonstrated that the abiotic conversion of extracellular NH_2_OH contributes to N_2_O emission during aerobic ammonia oxidation [24].

Directly converting NH_2_OH to N_2_O or combining NO with NH_2_OH thus obtaining N_2_O, the anaerobic NH_2_OH detoxification pathway catalyzed by cytochrome P460 (CytL) in most AOB is also a significant source of N_2_O [25]. CytL can oxide 2 equivalents of NH_2_OH and 4 oxidizing equivalents to 1 equivalent of N_2_O under anoxic conditions [25]. Alternatively, it can reduce NO to N_2_O in the presence of NH_2_OH [25]. CytL is used by AOB to detoxify NH_2_OH and NO, such that AOB can abundantly emit N_2_O from hydroxylamine oxidation under anaerobic conditions, thereby establishing a direct enzymatic link between nitrification and N_2_O production via NH_2_OH [25,26].

### 2.2. Nitrifier Denitrification

NO and NH_2_OH are two of the precursors of N_2_O emission during denitrification by nitrifiers at low DO conditions [26]. During nitrifier denitrification, NO_2_^−^ is reduced by nitrite reductases (NIR) to NO, which is further reduced to N_2_O through nitric oxide reductases (NOR) produced by nitrifiers. As such, this process is also a source of N_2_O in anammox-driven WWTPs [27], with Chen et al. [28] claiming that it produced 73% of N_2_O in a one-stage PNA reactor.

NO is a highly reactive and potent toxic molecule that can be converted to N_2_O by the enzyme NOR in AOB, AOA, and comammox [29]. Most AOB have NOR-encoding genes (*norB* and/or *norC*) to detoxicate NO [30]. Previously, despite the presence of *nir* genes in almost all AOA genomes, AOA were believed to be incapable of N_2_O production through nitrifier denitrification as they lack NOR [22,31,32]. However, a recent study found that cytochrome P450NOR in AOA can act as NOR leading to the production of N_2_O via nitrifier denitrification at low pH under aerobic conditions [33]. This notion is supported by the general N_2_O production pathway [2NO + NAD(P)H + H^+^ → N_2_O + H_2_O + NAD(P)^+^] by the enzyme P450NOR in archaea denitrification [34]. Putative cytochrome P450-encoding genes were found not only in the genomes of AOA but also in the genomes of AOB and comammox [34,35]. However, ^15^N isotope tracer analysis revealed that the comammox strain of *N. inopinata* cannot denitrify NO to N_2_O and thus emit N_2_O at a level that is comparable to that of AOA (much lower than that of AOB) under varying oxygen regimes, suggesting that N_2_O formed by *N. inopinata* mainly originates from the abiotic conversion of NH_2_OH [23]. Considering that P450NOR is not thought to be involved in energy conservation in fungal denitrifiers [36] and the contribution of the haem copper oxidase family (qNOR and cNOR) likely surpasses that of other NOR types due to their predominant roles in denitrification [37], it was suggested that AOA and comammox have weak N_2_O emission potential under anoxic conditions [23,31]. Although the N_2_O yield is significantly higher in nitrifier denitrification catalyzed by P450nor in AOA under aerobic conditions at low pH, it is still lower than that obtained by nitrifier denitrification catalyzed by NOR and hydroxylamine oxidation catalyzed by CytL in AOB under low-oxygen conditions [33]. Therefore, AOB are the dominant N_2_O producers during the partial nitrification process [23,24,38].

### 2.3. Heterotrophic Denitrification

Heterotrophic denitrification is one of the main nitrogen removal pathways based on the reduction of NO to N_2_O in wastewater by denitrifiers under anaerobic conditions, which begins by reducing NO_3_^−^ to NO_2_^−^ by nitrate reductases [27]. The produced NO_2_ is then reduced to NO through either haem-containing (cd1-NIR, *nirS*) or copper-containing (Cu-NIR, *nirK*) nitrite reductases, which is further reduced to N_2_O through NOR [27]. N_2_O is an intermediate product during denitrification, and part of N_2_O can escape from the cell before the final reduction to N_2_, resulting in N_2_O emission [39]. Microbial N_2_O reduction to N_2_ is the main sink of this powerful GHG, which is catalyzed by the enzyme nitrous oxide reductase (NOS) [27]. It is becoming apparent that complete denitrifiers that reduce NO_3_^−^ all the way to N_2_ are the exception and that many denitrifiers, called incomplete denitrifiers, lack NOR or NOS and directly use NO or N_2_O as the end product [27].

In microbial processes, NO is generated via NO_2_^−^ reduction catalyzed by NirS and NirK, which are functionally equivalent but structurally divergent [40]. The genes for these two enzymes rarely co-occur in the genome of denitrifiers [41,42]. Changes in the composition and diversity of the denitrifier community and differences in habitat preferences indicate a niche differentiation process leading to *nirK*- and *nirS*-type denitrifiers [42,43,44]. A clear separation of *nirS* and *nirK* communities was observed in saline and non-saline environments, with *nirS* communities dominating in marine environments [42]. Interestingly, the *nosZ* gene has a higher frequency of co-occurrence with *nirS* than with *nirK*, and *nirS* usually co-occur with *nor* [44]. Under favorable conditions, *nirS*-type denitrifiers are more likely to be capable of complete denitrification and usually contribute less to N_2_O emission than *nirK*-type denitrifiers [44]. The non-random patterns of *nir*/*nor*/*nos* gene occurrence [44] are important in determining the genetic N_2_O production potential in wastewater treatment systems and illustrate the importance of the microbial community structure for biotic N_2_O emission.

### 2.4. NosZ-Dominated N_2_O Sink

N_2_O-reducing microorganisms can reduce N_2_O to N_2_; therefore, their abundance and activity can strongly affect the net N_2_O emission from WWTPs. N_2_O degradation is catalyzed by members of either NosZ clade I or NosZ clade II. They can be distinguished by the signal peptide motif of twin-arginine translocation (Tat) or secretory (Sec) proteins, which govern the secretion pathway for N_2_O translocation across the cell membrane [45,46]. Clade II NosZ is characterized by a much broader diversity of microorganisms than Clade I NosZ. About 30% of Clade II NosZ lack a complete denitrification capability and are termed *nosZ* II non-denitrifiers [44,47]. The *nosZ* II non-denitrifiers are regarded as N_2_O reducers, as they lack other denitrifying enzymes that specifically consume N_2_O [16,44]. Hence, increasing the diversity and abundance of *nosZ* II-type non-denitrifiers could help N_2_O reduction in wastewater treatment systems [47,48]. Therefore, the community structure and regulatory mechanisms of *nosZ* II non-denitrifiers in anammox-driven wastewater treatment systems associated with N_2_O emission mitigation deserved more attention in future studies.

It is noteworthy that most studies attempting to characterize *nosZ* gene diversity using DNA-based PCR approaches only focused on Clade I *nosZ*, while the abundance and diversity of Clade II *nosZ* are underestimated [45,46]. The high diversity of Clade II NosZ makes it impossible to design a universal primer set that can effectively amplify all *nosZ* genes in this clade [49]. The Clade II *nosZ* community has yet to be thoroughly investigated, and characterizing its contributions to N_2_O consumption will significantly enhance our understanding of N_2_O emission in wastewater treatment.

## 3. N_2_O Emission Rate and Influence Factors

The N_2_O emission rate (0.057–2.3% of the total nitrogen load) varies substantially among different anammox-driven reactors (Table 1). The N_2_O emission rates are even higher in some anammox-driven reactors than in conventional nitrification/denitrification nitrogen removal systems (0.1–0.58% of the total nitrogen load) [50,51]. The high N_2_O emission rate is a major obstacle to the sustainable application of anammox systems for wastewater treatment. Factors such as DO, NH_4_^+^, and NO_2_^−^ concentrations, chemical oxygen demand (COD), and the presence of floc could significantly influence N_2_O emission by impacting the microbial communities and their activity in anammox-driven nitrogen removal systems.

### 3.1. Dissolved Oxygen

DO is a crucial operation parameter in anammox-processing systems. Maintaining a relatively low oxygen supply is suggested for PNA reactors to achieve partial nitrification by limiting oxygen availability to AOB [28]. As most NOB in wastewater treatment systems have low oxygen affinity, a low level of DO could inhibit nitrite oxidation by suppressing the activity of NOB [57,58]. However, a low level of DO could also stimulate N_2_O emission through heterotrophic denitrification and nitrifier denitrification in PNA systems [14,25,26]. A high oxygen supply not only promotes the nitrification process thus producing NO_2_^−^ rather than NO_3_^−^ and indirectly yielding N_2_O through hydroxylamine oxidation [13], but also suppresses the activity of anammox due to oxygen inhibition and NO_2_^−^ competition with NOB [59]. Balancing all factors, it is recommended that the oxygen concentration in anammox-driven nitrogen removal systems be kept at a low level to achieve partial nitrification and reduce N_2_O emission.

### 3.2. NH_4_^+^ and NO_2_^−^ Concentrations

The concentrations of NH_4_^+^ and NO_2_^−^ could significantly affect the level of N_2_O emission during wastewater treatment [51]. NH_4_^+^ can indirectly affect N_2_O emission through hydroxylamine oxidation or directly promote NO_2_^−^ production through nitrification [13]. A high NH_4_^+^ influx promotes NH_2_OH production and results in NH_2_OH accumulation, and part of NH_2_OH could leak out of the cell and enhance N_2_O emission during nitrification [52]. NO_2_^−^ is known to increase N_2_O emission through three main N_2_O production processes during wastewater treatment, i.e., hydroxylamine oxidation, nitrifier denitrification, and heterotrophic denitrification [60]. The presence of NO_2_^−^ not only offers a reactant for hybrid N_2_O formation from NH_2_OH via hydroxylamine oxidation but also delays the overall NH_2_OH abiotic decay, further stimulating the conversion of NH_2_OH to N_2_O [24]. Furthermore, NO_2_^−^ could increase N_2_O emission by inhibiting the N_2_O consumption activities of *nosZ*-containing denitrifiers [14]. Therefore, the concentration of NH_4_^+^ and NO_2_^−^ in anammox-driven nitrogen removal systems should be cautiously controlled to mitigate N_2_O emission.

### 3.3. Organics Availability

The positive effect of organic carbon on N_2_O mitigation has been reported in different reactors [17,59], with the addition of organics significantly reducing N_2_O emission (COD/N = 1) [52] and improving nitrogen removal efficiency (COD/N = 1.4) [61]. The presence of organic carbon provides energy to the growth of denitrifiers and boosts N_2_O consumption by easing the carbon limitation of N_2_O reduction to N_2_, which is the last step of denitrification [52]. The enhancement of anammox performance for wastewater treatment by the addition of a small amount of acetate has been reported [62,63], contributing to a reduction in metabolic energy cost for the entire community under a low C/N ratio [63].

It is noteworthy that N_2_O emission is enhanced by NO_2_^−^ accumulation from partial nitrification under low organics availability conditions [64,65]. Electron competition between *nosZ*-containing and other denitrifiers could be stimulated by low influent organics under high NO_2_^−^ conditions, such that N_2_O reduction by *nosZ*-containing denitrifiers could be inhibited [52,64,66,67,68,69]. High concentrations of organics could suppress anammox activity in anammox-driven systems [52,70], likely due to the competition between anammox bacteria and heterotrophic denitrifiers [52,70,71,72]. Additionally, denitrifiers in the presence of organic could increase N_2_O emission by affecting the number of flocs and filamentary structures around the anammox granules [52,73]. The variations in granule morphology could further affect N_2_O emissions due to DO fluctuation [52].

### 3.4. Flocs Formation

Flocs are present in all types of granular sludge reactors and suspended sludge reactors [74,75,76]. It was reported that flocs, which constitute only ~10% of the total biomass, contributed to 60% of the total N_2_O emission from a high-rate anammox granular sludge reactor [53]. The presence of small amounts of flocs has a non-negligible impact on nitrogen removal and N_2_O emission in anammox granule systems [77]. The abundance of *nirS* was shown to much greater than that of *nor* in both granules and flocs, which resulted in transient NO accumulation in the anammox reactor [53]. Flocs are associated with a high oxygen penetration depth, resulting in a relatively low abundance of anammox bacteria compared to AOB [42], while granules contain a large number of anammox bacteria at anoxic zoon, which could rapidly eliminate NO from other microorganisms [53]. The anammox bacteria are favored in relatively large granules [75,78,79]. The abundant NO-dependent anammox bacteria in granules could rapidly consume NO without the production of N_2_O (Figure 2), which suggests that anammox is a net NO consumption process associated with N_2_O emission mitigation in anammox granules [52,79,80]. Thus, this may explain why flocs are a significant source of N_2_O, due to NO accumulation (Figure 2).

In the nitrification/denitrification activated sludge system, it was reported that large flocs (>200 μm), in which heterotrophic denitrification that led to the generation of N_2_O was conducted by denitrifiers, showed higher N_2_O generation rates than small flocs (<100 μm) [42]. Denitrifiers usually coexist with anammox bacteria under anoxic or anaerobic conditions in anammox-driven wastewater treatment systems [81,82]. However, the contribution of denitrifiers in anammox granule has not been demonstrated. Nonetheless, anammox bacteria compete with denitrifiers for NO_2_^−^ in anammox-processing systems [71], so denitrifiers might not be as important as they are in nitrification/denitrification systems.

## 4. N_2_O Mitigation Strategies

Based on previous analyses, N_2_O emission in anammox-driven WWTPs can be reduced by (i) lowering DO concentrations (controlling the nitrification process), (ii) adopting intermittent aeration (motivating N_2_O degradation), (iii) reducing NO_2_^−^ concentration (controlling the nitrification and denitrification processes), and (ⅳ) increasing the C/N ratio (controlling the heterotrophic denitrification process). Additionally, regulating the microbial community composition, such as eliminating N_2_O producers and increasing N_2_O consumers, can be a potential N_2_O emission mitigation strategy.

### 4.1. Operational Parameters Control

As shown in Table 1, DO control is the most frequently implemented strategy to mitigate N_2_O emission in anammox-processing systems. This strategy has been implemented in a full-scale conventional nitrification/denitrification WWTP, resulting in a 35% reduction of N_2_O production via the hydroxylamine oxidation pathway [51]. Instead of continuous aeration, intermittent aeration could reduce N_2_O emission by allowing heterotrophic denitrifiers to consume N_2_O and/or N_2_O precursors (NO, NO_2_^−^) during anaerobic periods, and hence is the most widely adopted approach. It was also suggested that NO_2_^−^ can be maintained at relatively low levels using a recycling pump to avert N_2_O accumulation [54], especially under limited organics conditions (low C/N rate) [64,65]. It was demonstrated that a high NO_2_^−^ concentration could stimulate N_2_O emission from nitrifier denitrification and heterotrophic denitrification processes and likely inhibit N_2_O reduction carried out by *nosZ*-containing denitrifiers [13,55]. The positive effect of a high NO_2_^−^ concentration on N_2_O emission during wastewater treatment could be mitigated by the addition of organic carbon, reducing NO_2_^−^ influence and maintaining a neutral pH [52].

### 4.2. Microbial Community Structure

The microbial community structure of activated sludge in WWTPs determines the nitrogen removal ability and the N_2_O emission potency [53]. Ammonia oxidizers, which provide anammox bacteria with NO_2_^−^ by partly oxidizing NH_4_^+^, are essential for nitrogen removal in anammox-processing systems. However, aerobic ammonia oxidation is usually accompanied by N_2_O production via hydroxylamine oxidation and nitrifier denitrification [83]. AOB are deemed a significant source of N_2_O emissions in anammox-driven systems [13,14,54], but the newly discovered comammox organisms have relatively low N_2_O emission potential under anoxic conditions due to the lack of NO reduction enzymes [23]. Comammox organisms could outperform AOB in low-DO reaction tanks [82,84,85], ndicating that comammox bacteria are better substitutes for AOB for anammox-driven reactors.

Considering that nitrifier-enriched flocs are a significant source of N_2_O emission, the regular elimination of flocs from anammox granule systems is an effective way to mitigate N_2_O emission [52]. It was reported that removing 15% of flocs (2.8% of total biomass) can result in a significant decrease in N_2_O emission under constant DO conditions [52]. It should be noted that floc removal at a constant airflow rate could lead to DO fluctuations because of the reduced total oxygen consumption from nitrifiers [49,76]. Although part of AOB biomass is removed with the floc, a high DO concentration can stimulate hydroxylamine oxidation and hence generate more N_2_O. Therefore, a lower airflow rate is required during floc removal to maintain constant DO levels and control N_2_O emission from hydroxylamine oxidation.

Incomplete denitrification is also a significant source of N_2_O emission from WWTPs. The abundance of *nir* genes can exceed that of *nosZ* by up to an order of magnitude in various environments [45]. Thus, bacterial community composition and the co-occurrence of *nirS*, *nirK*, and *nor* with *nosZ* are expected to have a significant influence on the genetic N_2_O emission potential from wastewater treatment systems. Additionally, selectively inoculating and increasing N_2_O-consuming *nosZ* II non-denitrifiers in anammox-driven WWTPs is a promising N_2_O mitigation option [44,47,86].

Besides, anammox bacteria can reduce N_2_O emission by effectively consuming the accumulated NO in activated sludge or granules [52,79,80]. Anammox bacteria biomass is more abundant in granules than in flocs in the anammox granule system [53] so that granules have generally lower N_2_O emission rates compared to flocs [79]. Consequently, anammox may be a potential microbial process in NO and N_2_O emission control during wastewater treatment [79,80]. Inoculation of mature sludge with highly active anammox granules is an effective way to rapidly enrich anammox pellets and achieve a stable anammox-driven nitrogen removal process in ammonium-rich conventional WWTPs [82,84], which will significantly reduce N_2_O emission from nitrogen removal.

## 5. Evaluation of N_2_O Mitigation Strategies

N_2_O emission prediction models are a useful tool for evaluating the proposed N_2_O mitigation strategies and their effects on nutrient removal performance in full-scale WWTPs. The models typically use elements including microbial N_2_O generation and reduction pathways, as well as influence factors to simulate the real N_2_O emission and appraise mitigation strategies (Figure 3).

Mathematical models have been successfully applied to evaluate N_2_O mitigation strategies by quantifying nitrogen removal in conventional full-scale WWTPs [37]. Among various published mathematical N_2_O models, the ASM2d-N_2_O model developed by Massara et al. [87], which is a kind of activated sludge model (ASM), has been widely used for assessing N_2_O emission from full-scale WWTPs [38,88,89]. Besides the classical mathematical models, novel machine learning methods, such as deep neural network (DNN) and long short-term memory (LSTM), have also been used for N_2_O emission prediction [90].

Mathematical models developed based on the biological metabolic mechanisms of N_2_O production and consumption can easily calibrate N_2_O-related reactions by applying specific reaction kinetics parameters [87,91,92]. However, this requires a deep understanding of the N_2_O emission mechanisms and of the specific liquid–gas transformation variables in different WWTPs. On the contrary, deep learning models mainly rely on operational datasets with correlative features of the WWTPs. Hybrid modeling concepts, integrating mathematical models and deep learning models, have been suggested for evaluating N_2_O mitigation strategies [90]. A hybrid model combining mechanistic (ASMs) with an LSTM-based deep learning model has been successfully and accurately used for modeling N_2_O emission in a full-scale WWTP, with relatively low data requirements [90]. Anammox-driven nitrogen removal technologies have been widely used for wastewater treatment, but to our best knowledge, the current models have not been used to evaluate N_2_O emission in full-scale anammox-driven WWTPs. To increase the sustainability of anammox in wastewater treatment, more efforts are needed to evaluate the effects of the abundance and activities of anammox organisms and the mitigation strategies on N_2_O production in anammox-driven WWTPs.

## 6. Conclusions and Implications

Biologically toxic N_2_O is considered the third most emitted GHG contributing to global warming, and its concentration in the atmosphere has been steadily increasing in recent years. N_2_O emission is still one of the downsides of anammox-driven wastewater treatment, which accounts for 0.057–2.3% of nitrogen loading in anammox-driven systems and 0.1–0.58% of nitrogen loading in traditional nitrogen removal systems. In anammox-driven wastewater treatment systems, N_2_O is produced through three pathways, i.e., hydroxylamine oxidation, nitrifier denitrification, and heterotrophic denitrification, and is reduced through the unique pathway of *nosZ*-dominated N_2_O degradation. Biological processes, operational conditions (e.g., NH_4_^+^, NO_2_^−^, DO, COD), and microbial communities can affect N_2_O emission.

Common N_2_O mitigation strategies for WWTPs include DO control, aeration control, NO_2_^−^ limitation, C/N ratio control, and flocs removal regulation. Nonetheless, other potential strategies deserve further investigations, These include (i) increasing the biomass and activity of anammox bacteria, which are net NO consumers; (ii) the inoculation of N_2_O-reducing organisms, such as *nosZ* II non-denitrifiers with high N_2_O-affinity; (iii) establishing a symbiotic association of low-N_2_O-yield comammox and anammox.

The feasibility and efficiency of the proposed mitigation strategies need to be verified and optimized by prediction models, such as mathematical models and deep learning models, in practical application. The development of high-throughput sequencing techniques and data analysis methods can elucidate the structure of the microbial community in WWTPs at high-resolution and low cost and can potentially uncover in great detail N_2_O production and consumption mechanisms by the major microorganisms present in WWTPs. Therefore, more omics studies are needed to extend our understanding of the metabolic mechanisms of N_2_O emission in anammox-driven WWTPs, which will help us find out and formulate effective N_2_O emission mitigation strategies.

## Figures and Tables

**Figure 1 life-12-00971-f001:**
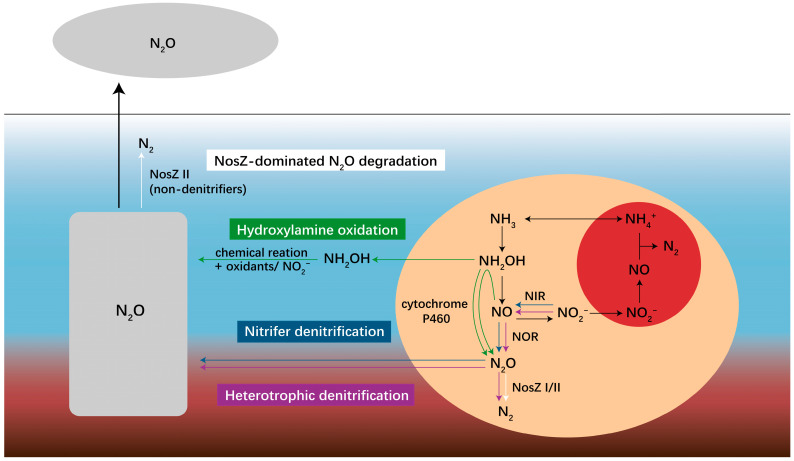
Schematic diagram illustrating the microbial pathways leading to N_2_O production (green, blue and purple boxes) and consumption (white box) in the anammox-driven reactor. The blue and red colors in the background represent wastewater and sludge, respectively, the red circle denotes the anammox reaction, and the orange circle denotes the nitrification and denitrification reactions driven by nitrifiers and denitrifiers.

**Figure 2 life-12-00971-f002:**
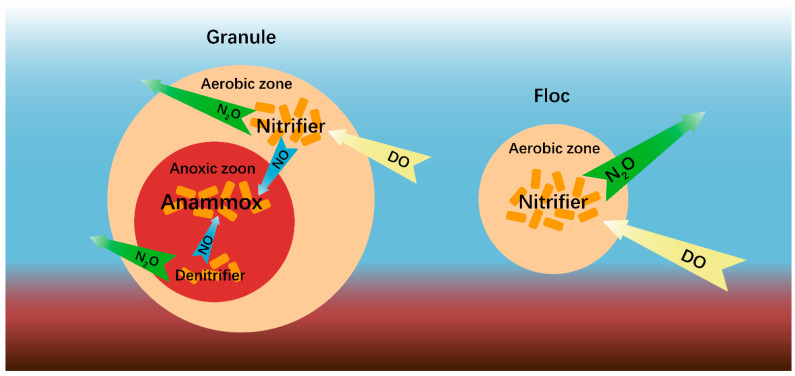
Flocs are a significant source of N_2_O emission in the anammox granule system. The blue and red colors in the background represent water and sludge, respectively. The yellow, blue, and green arrows indicate DO, NO, and N_2_O, while the red circle denotes the anoxic zoon containing anammox bacteria and denitrifiers, and the orange circle denotes the aerobic zoon containing nitrifiers. The number of deep orange square indicates the amount of nitrifiers, denitrifiers, and anammox bacteria in granule and floc.

**Figure 3 life-12-00971-f003:**
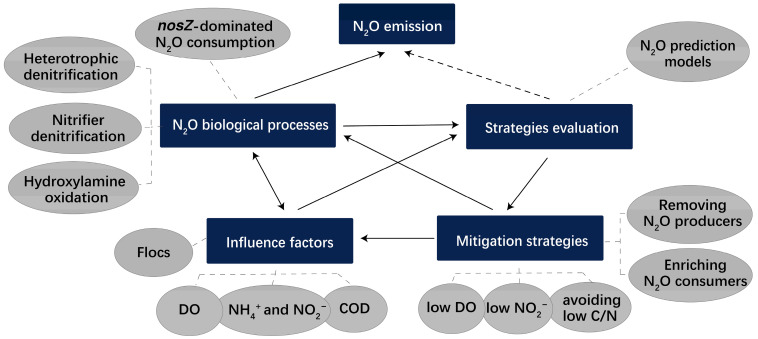
Schematic of strategies evaluation, mitigation strategies, influence factors, N_2_O biological processes, as well as N_2_O emission in anammox-driven WWTPs.

**Table 1 life-12-00971-t001:** Measured N_2_O emission flux and DO levels in different types of reactors. PNA, partial-nitrification/anammox, AMX, amammox.

Reactor	Strategies	DO (mg/L)	Nitrogen Removal Efficiency (%)	N_2_O Emission Rate (%) ^1^	Emission Factors	Reference
Lab-scale	one-stage PNA	<1	-	1 ^2^	DO, NH_4_^+^ and NO_2_^−^	[13]
	one-stage PNA	0.2−2.3	70.87 ± 1.36	0.004−0.11	Aeration control	[28]
	one-stage PNA	2	73.8 ± 4.1	1.0−4.1 ^3^	Influent organics, aeration control, flocs and NO_2_^−^	[52]
	AMX	≈0	86.7 ± 2.5	0.284	O_2_ and aggregate size	[47]
	AMX	<1	87.01	0.57 ± 0.07 ^3^	Flocs	[53]
	AMX	<0.5	>80	0.6−1.0 ^2^	NH_4_^+^	[54]
Full-scale	two-stage PNA	2.5	>90	1.7 (nitrification)-0.6 (anammox)	DO and NO_2_^−^	[14]
	one-stage PNA	<1	>90	0.4	DO	[50]
	one-stage PNA	0.5−1.5	>90	0.2−0.9 ^2^	DO	[55]
	one-stage PNA	0.5−1.5	81	0.35−1.33	Aeration control and the nitrogen loads	[56]

^1^ N_2_O-N of the total nitrogen load. ^2^ N_2_O/N^2^ yield of removed nitrogen. ^3^ N_2_O-N of the total nitrogen removal.

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
