# Peer review of "Nitrous Oxide Emission from Full-Scale Anammox-Driven Wastewater Treatment Systems"

_life, 2022, doi:10.3390/life12070971_

Round 1

Reviewer 1 Report

This study raises a significant point for the research community.  Which impact the GWP but the authors explain it all in words rather than comparing the existing studies. 

For example- The below point may assist the authors:

1- There are no tables found that describe the GHG effect on existing studies,

2- need to input comparative studies in terms of the table,

3- Authors mentioned fundamental things are more rather than putting experimental/modelling/practical existing studies,

Author Response

Reviewer #1: (Comments for the Author):

Title: Nitrous oxide emission from full-scale anammox-deriven wastewater treatment systems

Manuscript Number:

This study raises a significant point for the research community. Which impact the GWP but the authors explain it all in words rather than comparing the existing studies

For example- The below point may assist the authors:

1- There are no tables found that describe the GHG effect on existing studies,

2- need to input comparative studies in terms of the table,

3-Authors mentioned fundamental things are more rather than putting experimental/modelling/practical existing studies.

We thank the Reviewer for this encouraging assessment and constructive criticisms of our manuscript. As suggested, the GHG emission from diverse reactors on existing studies were summarized in Table 1. N2O emission rates from different systems were described in lines 195 to 203. More experimental/modelling/practical existing studies were given in Table 1 and lines 195 to 203, and we believe that fundamental things are also important to help readers to understand the basic knowledge of N2O emission during wastewater treatment.

Reviewer 2 Report

The present paper represent a semnificative work  regarding  the key factors influencing N2O  emission and mitigation strategies were discussed in detail, and areas in which further research urgently required were identified. 

Author Response

Reviewer #2: (Comments for the Author):

The present paper represent a semnificative work regarding the key factors influencing N2O emission and mitigation strategies were discussed in detail, and areas in which further research urgently required were identified. 

We thank the Reviewer for this positive and encouraging assessment of our manuscript.

Reviewer 3 Report

This paper presents the latest developments on the water treatment system based on energy efficient process anammox using new insights into N biotransformation involving metabolic pathways of hydroxylamine ,N2O, its elimination and functional microorganisms involved during the formation of process in treatment systems. Three main N2O production processes, namely hydroxylamine oxidation, nitrifier denitrification, and heterotrophic denitrification, and the unique N2O consumption process, namely nosZ-dominated N2O degradation, in the anammox-driven  wastewater treatment systems were summarized and discussed. The key factors influencing N2O  emission and mitigation strategies were discussed in detail, and areas in which further research urgently required were identified.

The treatment performance, enzymes NosZ and microbial communities were studied, nitrogen removal mechanism of wastewater was analyzed. The results showed that when the anammox N2O accounts for 0.057-2.3% of nitrogen loading in anammox-driven systems and 0.1-0.58% of  nitrogen loading in traditional nitrogen removal systems. However, removal rate could be given, too.

This study proposes the evaluation of wastewater treatment focusing on the simultaneous removal of nitrogen from wastewater and low N2O emissions.

not clearly understandable the novelty part of study.

- write in full all abbreiations after first appearance as WWTP in intro

Graphical abstract should not use abbraviations such as: CytL etc

Text just be justified. Three line table format should be used

What are the rates of TN removal that are described. Numeric results not needed to add to results.

Check the usage of supersciptions and subscriptions in your chemicals names.

Abstract missing any results on treatment through new performance was it successful or not, it have not shown. Efficiencies does not enough show process activities rather removal rate itself.

Check the statements and give proportions in treatments efficiencies within.

 On Fig. 2 you mention anoxic zoon- it should be "zone"

After: “ Anammox bacteria provides a shortcut in the nitrogen cycle by direct transforming ammonium (NH4 +) and nitrite (NO2 - ) to nitrogen gas (N2) [5], rendering it more efficient and cost-effective than the conventional nitrification/denitrification process „ Considering the limitation of conventional sewage treatment systems, the combination of biological processes (Moving bed bio film reactor – MBBR MFC stands as a  promising and viable option for water treatment and energy generation. Literature has shown different environmental treatments to be solved for more economic way, which could be shown: https://doi.org/10.1016/j.scitotenv.2021.149133, https://doi.org/10.1016/j.egypro.2011.09.002, https://doi.org/10.3390/w13030350

Under nitrifier denitrification AOA is included as important pathway, however, AOA are known to perform mostly ammonia oxidation by archaea, eleborate

Author Response

Reviewer #3: (Comments for the Author):

This paper presents the latest developments on the water treatment system based on energy efficient process anammox using new insights into N biotransformation involving metabolic pathways of hydroxylamine ,N2O, its elimination and functional microorganisms involved during the formation of process in treatment systems. Three main N2O production processes, namely hydroxylamine oxidation, nitrifier denitrification, and heterotrophic denitrification, and the unique N2O consumption process, namely nosZ-dominated N2O degradation, in the anammox-driven  wastewater treatment systems were summarized and discussed. The key factors influencing N2O emission and mitigation strategies were discussed in detail, and areas in which further research urgently required were identified.

The treatment performance, enzymes NosZ and microbial communities were studied, nitrogen removal mechanism of wastewater was analyzed. The results showed that when the anammox N2O accounts for 0.057-2.3% of nitrogen loading in anammox-driven systems and 0.1-0.58% of  nitrogen loading in traditional nitrogen removal systems. However, removal rate could be given, too. This study proposes the evaluation of wastewater treatment focusing on the simultaneous removal of nitrogen from wastewater and low N2O emissions.

We thank the Reviewer for this positive and encouraging assessment of our manuscript. The nitrogen removal rates were given as suggested (Table 1).

not clearly understandable the novelty part of study.

As suggested, the novel of this study was highlighted in lines 62-66.

- write in full all abbreiations after first appearance as WWTP in intro

As for the reviewer’s concern, the full descriptions of the abbreviations like WWTP etc were supplemented in the revised manuscript.

Graphical abstract should not use abbraviations such as: CytL etc

“CytL” was changed to “cytochrome P460”.

Text just be justified. Three line table format should be used

Three line table was used as suggested (Table 1).

What are the rates of TN removal that are described. Numeric results not needed to add to results.

The nitrogen removal rates were given as suggested (Table 1).

Check the usage of supersciptions and subscriptions in your chemicals names.

Chemicals names such as NH4+ and NO2- were carefully checked in the whole manuscript.

Abstract missing any results on treatment through new performance was it successful or not, it have not shown. Efficiencies does not enough show process activities rather removal rate itself.

Check the statements and give proportions in treatments efficiencies within.

Treatments efficiencies in the abstract were given as suggested (lines 12-14 and 37-39).

 On Fig. 2 you mention anoxic zoon- it should be "zone"

“zoon” was changed to “zone” (Fig. 2).

After: “ Anammox bacteria provides a shortcut in the nitrogen cycle by direct transforming ammonium (NH4 +) and nitrite (NO2 - ) to nitrogen gas (N2) [5], rendering it more efficient and cost-effective than the conventional nitrification/denitrification process „ Considering the limitation of conventional sewage treatment systems, the combination of biological processes (Moving bed bio film reactor – MBBR MFC stands as a promising and viable option for water treatment and energy generation. Literature has shown different environmental treatments to be solved for more economic way, which could be shown: https://doi.org/10.1016/j.scitotenv.2021.149133, https://doi.org/10.1016/j.egypro.2011.09.002, https://doi.org/10.3390/w13030350

More environmental treatments were given as “Considering the limitation of conventional wastewater treatment systems, the com-bination of biological processes such as sequencing batch reactor (SBR) stands as a promising and viable option for sewage treatment, with low cost, high efficiency and stability [8–10]” and the provided papers were cited in the modified version (line 44-47).

Under nitrifier denitrification AOA is included as important pathway, however, AOA are known to perform mostly ammonia oxidation by archaea, eleborate

N2O production by AOA in wastewater treatment was described (line 139-144). AOA and AOB are well-known to perform mostly ammonia oxidation through the partial nitrification process. Although nitrifier denitrification in AOA also contributes to N2O production, AOA are not major nitrifiers in wastewater treatment. As reported by Joo-Han et al. 2020, archaeal nitrification could be constrained by copper complexation with the high organic matter in wastewater treatment systems, which seriously limits the application of AOA in wastewater treatment.

Reviewer 4 Report

The review paper entitled ''Nitrous oxide emission from full-scale anammox-driven wastewater treatment systems'' aimed at evaluating N2O related factors in anammox driven wastewater treatment plants. The review paper provides a comprehensive survey of recent studies on the subject and a detailed discussion. The paper is well written and organized. I think it is acceptable without requiring any revision.

Author Response

Reviewer #4: (Comments for the Author):

The review paper entitled ''Nitrous oxide emission from full-scale anammox-driven wastewater treatment systems'' aimed at evaluating N2O related factors in anammox driven wastewater treatment plants. The review paper provides a comprehensive survey of recent studies on the subject and a detailed discussion. The paper is well written and organized. I think it is acceptable without requiring any revision.

We thank the Reviewer for this positive and encouraging assessment of our manuscript.

Round 2

Reviewer 1 Report

Dear Authors,

I have read your highlighted text and glad to agree your point of view. I am accepting this manuscript in this present version.

Thanks